# The Development of an Electron Pulse Dilation Photomultiplier Tube Diagnostic Instrument

**DOI:** 10.3390/s24237497

**Published:** 2024-11-24

**Authors:** Wenyong Fu, Chenman Hu, Ping Chen, Rongyan Zhou, Ling Li

**Affiliations:** 1Nanyang Institute of Technology, Nanyang 473004, China; wyf10071121@126.com (W.F.); 3162113@nyist.edu.cn (C.H.); chenpingszu@126.com (P.C.); zhoury619@163.com (R.Z.); 2College Department of Electronic and Information Engineering, Hengshui University, Hengshui 053000, China

**Keywords:** inertial confinement fusion, PD-PMT, temporal resolution, recompress, first collision

## Abstract

A new pulse-dilated photomultiplier tube (PD-PMT) with sub-20 ps temporal resolution and associated drivers have been developed for use detection and signal amplification in the inertial confinement fusion (ICF) community. The PD-PMT is coupled to a transmission line output in order to provide a continuous time history of the input signal. Electron pulse dilation provides high-speed detection capabilities by converting incoming signals into a free-electron cloud and manipulating the electron signal with electric and magnetic fields. This velocity dispersion is translated into temporal separation after the electrons transit into a drift space. The free electrons are then detected by using conventional time-resolved methods and the effective temporal resolution is improved about 12 times. In order to accurately obtain the actual device input signal, we experimentally investigated the relationship between microchannel plate (MCP) gain and electron energy during the first collision. We report the measurements with the PD-PMT, and the error source of the amplitude of the compressed signal is analyzed, which provides a reference for subsequent accurate construction.

## 1. Introduction

Ultrafast diagnostic equipment is an indispensable research tool in cutting-edge scientific research and technology [1,2,3]. Inertial confinement fusion (ICF) occurs under extreme conditions such as extremely short time (~10^−9^ s), extremely small space (0.1–1 mm), extremely high temperature (100 million degrees), and extremely high density (300 g/cm^3^). Therefore, it can only be detected using diagnostic equipment. This equipment will help in developing diagnostic techniques under extreme transient conditions, using more advanced diagnostic techniques to study previously unexplored scales of pressure, temperature, time, and space, providing strong technical support for more precise research on physical problems, and promoting progress in physical understanding [4,5]. X-ray measurement equipment used in ICF experiments includes photoconductivity detectors, thermoelectric detectors, and X-ray diode detectors. Out of these, vacuum photodiode detectors are preferred because of their fast response time, high reliability, and convenient sensitivity calibration. A reflective photodiode is a common type of vacuum photodiode detector that mainly uses X-rays to traverse the mesh anode, irradiate the cathode, and generate photoelectrons. The photoelectrons return to the anode after acceleration by a DC electric field, thus generating a pulse current in the output circuit to produce an electrical pulse with a time resolution of tens of ps. The photodiode used by the Lawrence National Laboratory in the United States has the fastest response time of about 50 ps, while that used in Shenguang III has a response time of about 55 ps [6,7]. Such conventional diagnostic techniques cannot meet the requirements of high temporal and spatial resolution and wide dynamic ranges. In ICF, implosion burn widths are typically 150 ps and are expected to drop as performance improves [8]. For adequate characterization of the dynamics near stagnation, detectors with a temporal response in the order of 10 ps are required. There are two main ways to improve the time resolution of the detection signal: one is to improve the time resolution of the detection equipment itself, and the other is to improve the time resolution of the detection device by using pulse dilation technology to broaden the detected signal in space. This limitation has motivated researchers to not only optimize the traditional ultrafine diagnostic technique but also explore and invest in its innovation. In particular, the application of the electron pulse dilation technique in PMT has significantly improved its time resolution capability [9,10,11,12]. This technique enables high-speed detection by converting incoming signals into free-electron clouds and manipulating electronic signals using electric and magnetic fields.

Prosser was the first to propose a technique for increasing the bandwidth of an electron detector by imparting velocity dispersion to the electron beam [13]. This technique converts short signals into longer time scales to slow down the probed signal, which improves the temporal resolution of the detector. Electron pulse dilation is a high-speed detection technique that can achieve a temporal resolution of several picoseconds. It is based on the principle of conversion of a signal of interest into a cloud of free electrons, which is accelerated into a vacuum drift space. The acceleration potential varies with time and causes the axial velocity to disperse in the electron cloud. This velocity dispersion is converted into temporal separation after the electrons pass through the drift space. Subsequently, the traditional time-resolved method is used to detect the free electrons, and the effective time resolution is “magnified” many times. In the past decade, different diagnostic instruments have been developed that use electronic pulse dilation technology to achieve temporal resolution in the range of 5–30 ps [14,15]. The driving force behind these development efforts is the demand for advanced diagnostics in high-density physics experiments carried out around the world.

Unlike the pulse dilation frame camera in which the MCPs are coupled to a CCD (Charge-coupled Device) readout for acquiring 2D time-gated images, the MCPs of the PD-PMT are coupled to a transmission line output in order to provide a continuous time history of the input signal. The objective of developing this PD-PMT is to increase the system bandwidth for a short time. Dilation not only temporally spreads the signal of the burn history but also reduces the signal amplitude by the dilation factor. As a result, there is a need for higher MCP gain factors in dilated measurements compared to undilated measurements. The dilated signal measured by a device should be compressed using a compression algorithm to provide the input signal. The actual device resolution time rate is obtained by recompressing the measured signal using three pieces of information: the electron beam dilated by magnification, the MCP gain factor, and the MCP impulse response function (IRF). It should be noted that the resolution of this measurement is not the final output of the device. In order to obtain the final device time resolution, pulse recompression is introduced into data processing, and therefore, the time resolution obtained is a semi-experimental result.

In this paper, by measuring the broadening ratio of the electron beam and the relative gain of the MCP, the input signal is obtained through compression of the data of the PD-PMT measurement results, and the source of the error is analyzed.

## 2. Detector Description

Figure 1 shows the experimental setup of the detection device PD-PMT. It consists of a photocathode, grid, magnetic lens, MCP, and charge collector. All components except the magnetic lens are located in a high-vacuum chamber with an air pressure of 3 × 10^−5^ torr. A gold photocathode with a microstrip structure is evaporated on quartz glass, where the width, length, and thickness of the microstrip lines are 8 mm, 35 mm, and 25 nm, respectively. A circular printed circuit board (PCB) is located outside the photocathode with two strip gradient output electrodes. Gold sheets are used to connect the microstrip gradient line of the PCB to the microstrip line of the photocathode. There is an approximate gap of 1 mm between the photocathode and the grid. The grid size is 20 lp/mm and the opening rate is 60%. The MCP and microchannel have diameters of 56 mm and 12 μm, respectively. The length-to-diameter ratio and the initial inclination angle of the channel are 40 and 6°, respectively, and the opening area is 60%. The distance between the grid and the MCP is about 490 mm, and a 0.5 mm thick PCB is deployed at 1 mm between the output surfaces of the MCP. The width, length, thickness, and impedance of the microstrip are about 2 mm, 200 mm, 3 μm, and 50 Ω, respectively. The output electrodes from the photocathode and collector are connected to four vacuum-sealed SMA connectors. The device is coated with a short magnetic lens.

The photocathode is loaded with a bias voltage of −3.2 kV. The experiment consists of irradiating the photocathode by using a femtosecond laser of wavelength equal to 266 nm through an aperture that has a diameter of about 1 mm. The emitted photoelectrons accelerate through the acceleration zone between the cathode grids and enter the drift zone. The external magnetic lens is used to magnetically focus the electron beam at the center of the MCP input surface. Bias voltages are applied at both ends of the MCP, and the incident electrons are multiplied by the MCP. The electrons departing from the MCP arrive at the microstrip line charge-receiving stage, resulting in signal generation and its recording by a 6 GHz bandwidth oscilloscope. The focusing of all electronic beams by the magnetic lens at the geometric center of the charge collector is ensured by first replacing the charge collector with an imaging system consisting of MCPs and phosphor screens. The current of the magnetic lens can be adjusted using this imaging system to ensure that the electronic image is imaged at 1:1 in the center of the MCP. The MCP input plane voltage and the output place voltage are −800 V and 0 V, respectively.

## 3. Experimental Results

In order to obtain the actual input signal from a dilated measurement, it is necessary to apply a recompression algorithm. The recompression of the signal requires three pieces of information, namely, the gain factor, the dilation factor, and the MCP-IRF.

### 3.1. Experimental Preparation

The laser pulse going to the PD-PMT is propagated through a Michelson interferometer, resulting in two light pulses of adjustable temporal separation. By adjusting the Michelson-type interferometer, the pulse spacing is 50 ps.

In the early days, laser triggered spark gaps were used to generate high-voltage slope electric pulses [16]. However, the triggering voltage requires several thousands of volts, resulting in large triggering shaking and a large system volume.

At present, the main ones that are widely used are thyristors, photoconductive devices, and avalanche transistors. An avalanche transistor pulse circuit can output thousands of volts of voltage, and has advantages of a low trigger voltage, little shaking, a fast rising edge (less than 1 ns), low power consumption, and a small size; avalanche transistors can be flexibly used in the circuit, transistors can be connected in series to obtain high voltage pulses, and parallel connection can obtain a high current output. So, the use of avalanche transistors as switching elements is the most widespread [17].

In order to improve the amplitude and ultrafast rising edge of picosecond driving electric pulse output, the usual method is to connect multiple avalanche tubes in series or in the form of a Marx pulse generator. The Marx pulse generator has the advantage of having a relatively low power supply voltage, but its distributed capacitance is relatively large, which affects the rise time of the output pulse. The relatively large distributed capacitance also has a significant impact on the stable operation of the circuit, especially in high-voltage-output situations; it will easily cause breakdown damage to the avalanche transistor. The series connection of avalanche transistors has a relatively small distributed capacitance, which helps to shorten the leading time of pulses. However, when using a large number of avalanche transistors, it requires a relatively high-DC high-voltage power supply, which makes it easy to generate high-voltage discharge and ignition in the circuit. The combination of these two methods is an ideal design scheme for high-voltage pulse circuits.

In order to ensure the steepness of the rising edge, the tandem avalanche tube must have a high degree of consistency in terms of breakdown voltage and trigger delay. During use, we noticed that the breakdown voltage value of this avalanche tube is quite different, and the creep of the breakdown voltage is also relatively large during the power-up test. These problems will directly affect the stability of pulse output amplitude and trigger delay, so the circuit cannot work in a stable state. In order to solve these problems, we carried out long-term aging experiments on these avalanche tubes, and selected the tubes with good breakdown voltage consistency, small breakdown voltage creep and trigger delay changes, and fast on-time ability.

The high-voltage dilation pulse generated by the avalanche tube circuit is loaded onto the microstrip cathode through an impedance gradient line [18,19]. The pulse waveform is shown in Figure 2, with a time of approximately 180 ps from point A to point B, a voltage of approximately 540 V, and a slope of approximately 3.0 V/ps.

In the limit of a small accelerating gap, the temporal magnification M can be approximated by
(1)$$M\approx 1+\frac{\tau_{d}}{2}\frac{\dot{\varphi }}{\varphi },$$
where $\tau_{d}$ is the electron drift time and φ(t) is the accelerating potential.

Under the conditions of a photocathode loading bias voltage of −3.2 kV, the slope of the dilating pulse is 3.0 V/ps, the drift zone length is 490 mm, and the widening ratio is about 8.9 to 10.9 times greater when the laser pulse is synchronized between point A and point C of the broadening pulse (with a distance of about 70 ps between point C and point B, ensuring that the broadening pulse changes linearly when the electron beam passes between the cathode and the grid).

### 3.2. Measurement of MCP Gain for First Collision with Different Electron Energies

In order to obtain the actual burn history from a dilated measurement, both the time and amplitude of the deconvolved measured signal need to be rescaled. The signal is recompressed by dividing the time between bins by the dilation factor and multiplying the amplitude by the dilation factor and then dividing it by the relative gain factor. The gain of electrons traversing the MCP is a necessary condition for recompression, and the gain error caused by the different electron energies during the first collision cannot be ignored.

The MCP was first developed by Goodrich and Wiley [20]. Since then, it has been widely used for measuring weak signals due to its high electronic gain. The MCP is made up of millions of regular arrays of tiny hollow glass tube channels obtained from fusion glass. Each channel forms an independent continuous dynode multiplier unit as its inner wall is covered with a dynode multiplier layer. Figure 3 shows the working principle of the MPC.

The incident electrons incident at the low-voltage end of the channel impinge on the inner wall of the channel to generate secondary electrons which are transmitted forward along the channel due to the acceleration caused by the electric field. They again collide with the inner wall of the channel, generating new secondary electrons, and the process is repeated until they leave through the exit end of the channel. This results in electron multiplication, where the electron multiplication function is based on the secondary electron emission effect.

The MCP adopts a three-layer metal coating. The bottom lining material is Cr, the middle layer is Cu, and the cathode material is Au. Due to its excellent bonding properties, Cr can effectively bond the metal coating with the lead glass substrate. Cu has extremely low electrical resistivity and excellent conductivity, while using Au plating on the outermost layer is beneficial due to its good quantum efficiency, stability when exposed to air, and small dispersion of the emitted photoelectron energy. These conductive coatings are formed by vapor deposition, with the input surface covered by normal deposition at 60° and the output surface covered by the same conductive layer deposited by normal deposition at 45°.

The gain of the electrons through the MCP is a necessary condition for the pulse recompression algorithm. The MCP gain at different bias voltages can be provided by the manufacturer. However, the energy of the electron beam that reaches the input surface of the MCP gradually decreases for the PD-PMT equipment. Therefore, it is necessary to consider the gain caused by different energies of the electrons at the time of the first collision. The energy of the electrons that collide with the MCP for the first time will directly affect its gain, detection efficiency, and noise factor. H. Geppert-Kleinrath compensated for the energy variation of the electron beam by loading pulses on the microstrip line of the MCP input plane [21]. Fu is compressed and reconstructs the time coordinates from the output pulse to the input pulse [22]. In this paper, an experimental study of the relationship between the electron gain of the MCP in the PD-PMT and the electron energy at the time of the first collision is carried out. This study provides a reference for the gain factor of one of the three conditions in the recompression algorithm of PD-PMT measurement results.

Figure 4 shows the experimental device for studying the relationship between the MCP gain and electron energy during the first collision. A third harmonic of a Quantronix Integra-C Ti–sapphire laser system is used. The laser outputs two laser beams with wavelengths of 266 and 800 nm. The 266 nm laser with a 130 fs pulse width is used to illuminate the photocathode to create photoelectrons. The 800 nm laser beam illuminates the p-i-n detector to generate a trigger signal.

This experimental setup is used to measure the relationship between the MCP gain and the energy of the electrons during the first collision with two methods. The first one involves fixing the bias voltage at both ends of the MCP, changing the bias voltage of the photocathode, and measuring the gain of the MCP under different input electron energies. The second method involves fixing the bias voltage of the photocathode and changing the input voltage of the MCP, while maintaining a constant difference between the bias voltages at the two ends of the MCP. In this case, the voltage at the output surface of the MCP must change. There is an electric field change between the MCP output surface and the charge-accepting microstrip line. We know that Ramo’s theorem gives the following formula for calculating the induced current [23]:(2)$$i=E_{v}qv$$
where i is the instantaneous current received by the given electrode due to a single electron’s motion, q is the charge of the electron, v is its instantaneous velocity, and $E_{v}$ is the electric field component in the direction of v that would exist at the electron’s instantaneous position under the following scenarios: electrons removed, the given electrode raised to a unit potential, or all other conductors grounded. Equation (2) involves the usual assumptions that the currents induced due to magnetic effects are negligible and the electrostatic field propagates instantaneously.

The initial value of the cathode bias voltage is −3000 V. The interval between each measurement is equal to 100 V, and the magnetic lens current is adjusted so that the laser image is imaged at 1:1 in the center of the MCP. The corresponding magnetic lens current is recorded. Subsequently, the phosphor screens are moved away and the collector is implemented. The above experimental steps are repeated by changing the bias voltage of the photocathode and the current of the magnetic lens. The oscilloscope can be used to obtain the corresponding electrical signal curves after MCP gaining under a series of different electron energies. Energy measuring devices are placed in the beam path to measure and account for shot-to-shot laser energy variations. Figure 5 shows the signal output of the charge collection microstrip line with a cathode bias voltage of −3000 V after being gained by the MCP.

As Figure 5 shows, the current transmission in the circuit ends at one time from $t_{0}$ to $t_{1}$. As the collected charge is obtained by the integration of current over time, and the current is equal to the voltage divided by the resistance, the total charge is written as follows:(3)$$Q_{0}=\int_{t_{0}}^{t_{1}}Idt=\int_{t_{0}}^{t_{1}}\frac{U}{R}dt$$
where $Q_{0}$ is the amount of charge collected by the charge collector, R is the external matching resistance, and U is the voltage value given by y-axis values of the curve collected by the oscilloscope. Therefore, the product of the pulse amplitude and the full width at half maxima (FWHM) can be used to represent the amount of charge collected. The collected charge is normalized between a cathode bias voltage ranging from −3.0 kV to −1.1 kV, and the corresponding results are shown in Figure 6.

The energy corresponding to the first collision between the electron beam and the MCP is related to the cathode bias voltage, the dilation pulse shape, the synchronous position of the laser and the pulse, and the voltage of the MCP input surface. The relationship between the MCP gain and the electron energy at the time of the first collision can be used to remove the gain from the experimental results and restore the input signal.

### 3.3. Signal Broadening Ratio Measurement

There are two methods that can be used to measure the aspect ratio of a signal. First, the broadening ratio of a signal can be calculated based on the peak-to-peak spacing of the broadened output signal $d_{dilated}$ and the peak-to-peak spacing of the un-broadened output signal $d_{undilated}$:(4)$$\lambda =\frac{d_{dilated}}{d_{undilated}}$$

Second, since dilation also delays the measured signal, a shift in relative timing between the PC high-voltage ramp and the arrival of the dilated signal is used to calculate dilation factors. We calculate the broadening ratio by utilizing the significant shift in the peak value of the broadening signal caused by the slight movement of the broadening pulse during the synchronization process between the laser and the broadening pulse:(5)$$\lambda =1-\frac{\Delta t_{peak}}{\Delta t_{ramp}}$$

Due to the small emission interval between two initial electron beam pulses, accurate measurement is difficult. Therefore, the accuracy of the results obtained using the method introduced in Equation (4) is relatively low. However, when using this method, the broadened signal peak-to-peak value is calculated from the same light source in one go, so its accuracy is relatively high. The second method for calculating the broadening ratio has a relatively high accuracy, as it is independent of the measurement uncertainty of the initial interval pulse. Nevertheless, the large error caused by the jitter phenomenon between the broadening pulses triggered by different signals greatly reduces the accuracy of this method. Currently, research is being conducted on how to reduce the jitter between the broadening pulses. This experiment uses the method introduced in Equation (3) to calculate the broadening ratio.

At a cathode bias voltage of −3.2 kV, the MCP input plane voltage and the output place voltage are −800 V and 0 V, respectively. Loading the broadening pulse at the photocathode, the synchronous laser pulse can obtain a broadening signal with different broadening magnifications at different positions of the A and C points of the broadening pulse. Figure 7 shows the widened pulses with peak-to-peak spacing of about 500 ps, and the input pulses are broadened about 10 times. The transit time spread of electrons through the MCP is about 40–50 ps [24], which is much shorter than the interval for the electron beam to reach the MCP, namely 500 ps. Therefore, the MCP could be capable of responding to input signals in a timely manner.

### 3.4. Measurement of MCP Gain for First Collision with Different Electron Energies

When the width of the input electronic signal (photoelectron beam) is much smaller than the width of the MCP pulse response function, it is necessary to perform a deconvolution calculation on the signal during the signal reconstruction process, fully considering the influence of the MCP pulse response function on the input signal. When the broadening ratio is around 10, the results obtained without deconvolution are basically consistent with the simulation results, indicating that the impact of convolution on the input signal is very small at this time. There is no need to calculate deconvolution when reconstructing the input signal, and the error of deconvolution is also small.

In order to obtain the actual burn history from a dilated measurement, both the time and amplitude of the deconvoluted measured signal need to be rescaled. The signal is recompressed by dividing the time between bins by the dilation factor and multiplying the amplitude by the dilation factor and then dividing it by the relative gain factor.

Since the dilation pulse used in the dilation time window is linear, the dilation rate corresponding to each time point within the window is nonlinear. In order to accurately determine the timeline of recompression during reconstruction, it is important to know where the signal is located in the recording window, as the widening rate varies over time. The time in the input signal can be obtained by using the dilation ratio to find the position of the input signal falling on the expansion pulse and the sampling information of the oscilloscope to calculate the expansion ratio and the peak of the first pulse at each sampling point. The synchronization position between the laser and the stretching pulse can be calculated by using the broadening ratio, and the time when the photoelectron escapes from the photocathode can be obtained. The voltage between the cathode and the grid varies linearly, so the time when the photoelectron beam exits the grid can be calculated. Combined with the bias voltage of the photocathode, the energy of the electron beam exiting the grid can be obtained. When the electron beam moves to the input surface of the MCP through the magnetic field confinement, the energy of the first collision between the electron beam and the MCP can be calculated from the voltage on the input surface and the energy of the electron beam out of the grid.

Figure 8 depicts the result of reconstructing the output signal; the recompression FWHM of the first dilation peak is 22 ps. The width of the second peak after compression reconstruction is about 19 ps. Compared with the FWHM of the pulse in Figure 3, it can be seen that its time resolution has been improved about 12 times. The amplitude ratio of the two pulses after decompression is about 1:0.88, which is considerably different from the ratio of 1:1 for the input pulse. After analysis, the following sources of error are identified:

During the pulse dilation experiments, the energy of electrons passing through the grid decreases due to the time-varying electric field between the cathode and the grid. Therefore, the current in the magnetic lens that makes the electron beam converge as 1:1 at the center of the MCP is also different. However, in the dynamic laboratory, the current of the magnetic lens is constant, which may result in some electrons not converging in the diameter range of 1 mm at the center point of the MCP. Consequently, the electrons cannot be completely collected. The dilation pulse causes the electric field to change unevenly in the space between the cathode and the grid, resulting in the deviation in the electron beam from the axis. This may affect the probability of electrons passing through the grid and their landing point on the MCP.The fluctuation in the incident light and electrons emitted by the photocathode can also cause error. Photoemission itself also exhibits random behavior, manifested as the probability that the quantum efficiency of the photocathode follows a Poisson distribution. The quantum noise of photoelectric emission is the main factor causing errors, while the noise of incident photons can be ignored.The MCP fluctuation includes the output electronic noise corresponding to the fluctuating number of photoelectrons input to the microchannel plate. This noise is caused by the fluctuation in the multi-stage secondary electron multiplication of the microchannel plate.

## 4. Conclusions and Outlook

Electronic pulse dilation technology has enabled the development of high-speed diagnostic instruments with picosecond time-resolution performance. The PD-PMT was developed with the aim of increasing the system bandwidth in a short period of time. The design configurations of this new instrument were described, including details regarding the photocathode elements and time-resolved electron detector backends. Magnetic imaging has been utilized to guide the electron signal through a drift region and prevent excess transverse spreading. In a PD-PMT experiment, in order to accurately reproduce its input signal, it is necessary to know where the signal falls in the recording window, as both the time magnification and the gain of the MCP backend vary with time. The experimental results show that the pulse dilation technique broadens the detected signal about 12 times, and the time resolution of the detector is improved to 19 ps. The experimental results show that the electron gain of the MCP increased with the increasing electron energy during the first collision. The amplitude ratio of the two pulses after decompression is about 1:0.88. Different aspects that introduced errors into the signal compression and reconstruction process were also discussed to provide a reference for further improving the measurement precision.

## Figures and Tables

**Figure 1 sensors-24-07497-f001:**
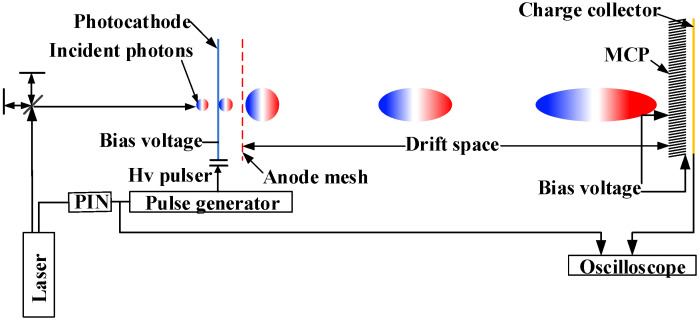
Schematic diagram of PD-PMT.

**Figure 2 sensors-24-07497-f002:**
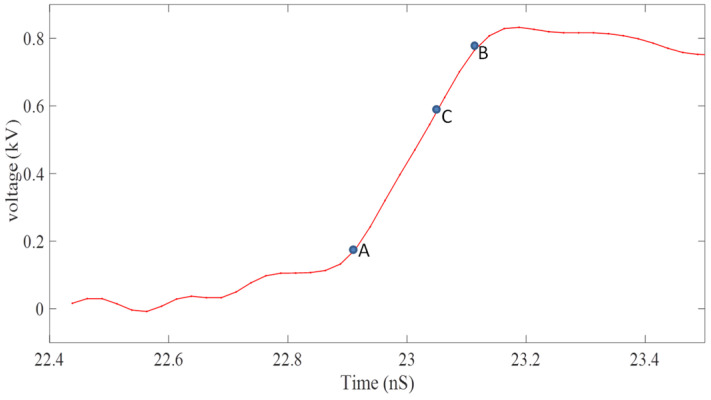
Broadening pulse waveform.

**Figure 3 sensors-24-07497-f003:**
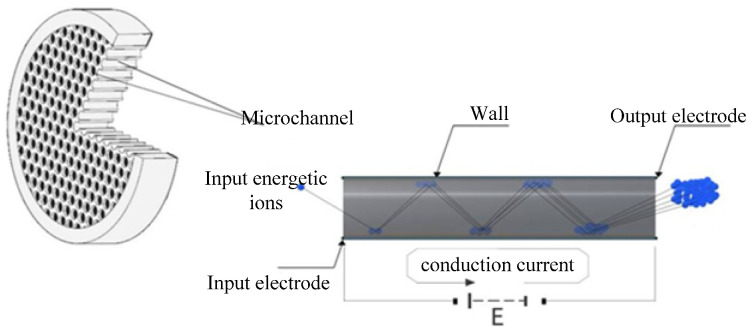
Schematic diagram of working principle of MCP.

**Figure 4 sensors-24-07497-f004:**
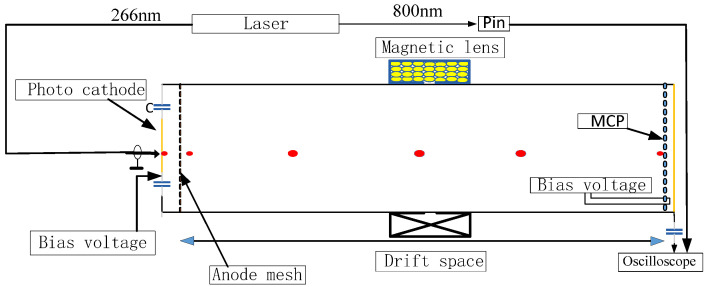
Schematic diagram of measuring MCP gain and electron energy during first collision.

**Figure 5 sensors-24-07497-f005:**
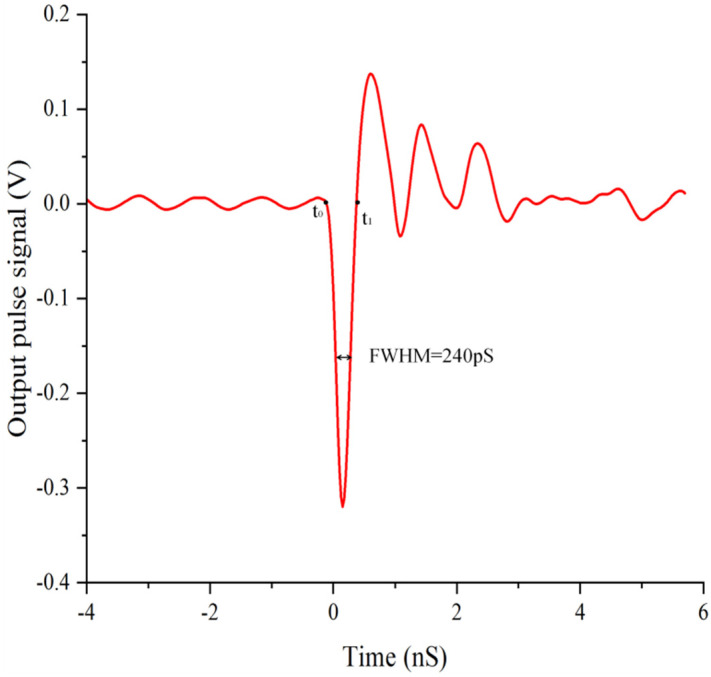
Signal acquired by oscilloscope at PC bias voltage of −3000 V.

**Figure 6 sensors-24-07497-f006:**
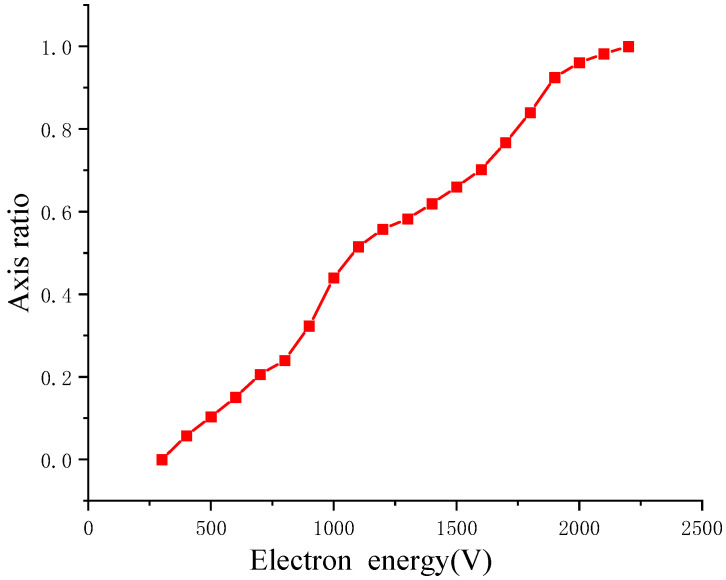
Normalized MCP gain vs. photoelectron energy in first collision.

**Figure 7 sensors-24-07497-f007:**
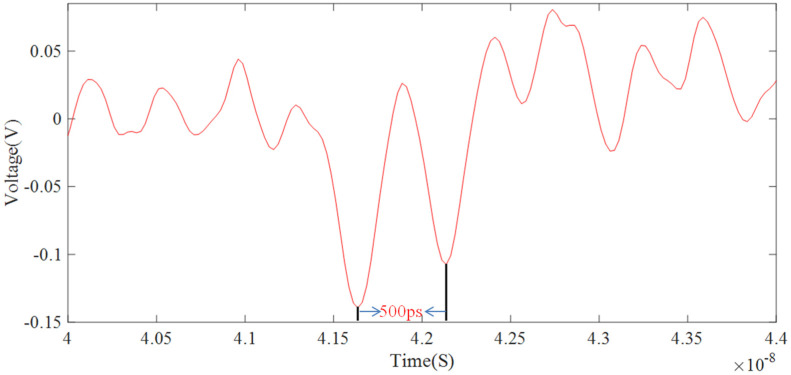
Dilated signals of PD-PMT.

**Figure 8 sensors-24-07497-f008:**
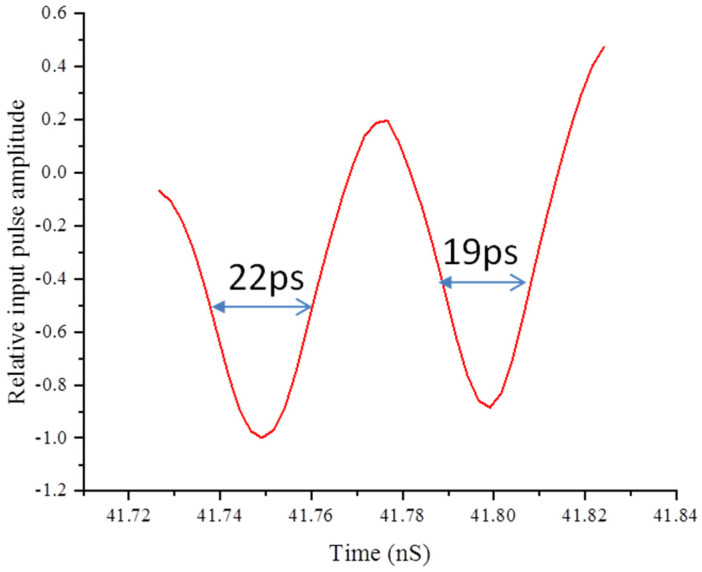
Recompression of dilated signals.

## Data Availability

The data presented in this study are available on request from the corresponding author.

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
