# Peer review of "The Development of an Electron Pulse Dilation Photomultiplier Tube Diagnostic Instrument"

_sensors, 2024, doi:10.3390/s24237497_

Round 1

Reviewer 1 Report

Comments and Suggestions for Authors

Figure 3 contains images with two very different lengh scales.  I think the including the relevant length scale would be very useful.

In figure 4 there is no explanation of the solid path labelled "laser".

I was confused by the reference to Kleinrath in reference 21, as I was unable to find a Kleinrath named in the author list or the article.  I was unable to check all references but "spot" checked this and a few other references.  It would be worth the authors time to check that each reference is referring to the appropriate publication.  I think "Fu is no treatment for gain in decompression[22]" is meaning "Fu has no treatment for gain in decompression[22]" but I found this section confusing.  Cleaning up the reference details and the English in this section would benefit the message significantly.

How is T0 and T1, shown in figure 5, chosen and does this change from measurement to measurement?  Are the measurements  dependent on this choice and how does this contribute to error?

Figure 6 shows Axis Ratio vs Electron Energy.  This is the only place where I was able to find Axis Ratio and was somewhat confused by what is plotted.  Was electron Energy extracted from bias voltage?  I would need an explanation of how this information was calculated from the information that was discussed in the text.

The text specified that the method described in equestion 3 is how the dilation broadening was measured but eqn 3 does not show how a broadening measurement is made?  Is this correct?

Therefore, the MCP could be 279 capable of responding to input signals in a timely manner.  This seems like an assertion most likely based on measurements.  Could you move from an assertion to a reference or to a measurement?

In the discussion of deconvolution there are adjectives used like "very small" or "small".  This needs to be quantified.  Something like <0.1% (or something) would provide the reader with some insight into the extent of the impact.

This could use more explanation.  I understand the importance but I'm not sure I understand the mechanism.

"Since the dilation pulse used in the dilation time window is linear, the dilation rate 295 corresponding to each time point within the window is nonlinear"

This seems redundant or I'm missing the point.

"However, in the dynamic laboratory, the current of the magnetic lens is constant that may result in some electrons not converging in the diameter range of 1 mm at the center point of the MCP. Consequently, the electrons cannot be completely collected.

  1. When the current of the magnetic lens remains constant, electron beams of different energies 323 converge at the center of the MCP, the incident angle of electrons entering the MCP will change, 324 resulting in a change in gain."

I expected there to be a curve of gain vs electron energy at the end of this analysis.  Perhaps this would be complicated by a dependency on time.  I guess this would need to be displayed in a 2D surface. 

The conclusion seems to be that the time resolution was improved to 19 ps and the gain is 1:0.88.  more description of this conclusion is needed.

Reviewer 2 Report

Comments and Suggestions for Authors

This research primarily investigates how to develop a photomultiplier tube (PD-PMT) diagnostic instrument capable of achieving sub-20 picosecond temporal resolution, specifically tailored for use in inertial confinement fusion (ICF) experiments. The focus is on leveraging electron pulse-dilation technology to enhance the detection capabilities of the PD-PMT, allowing for precise measurements under the extreme conditions typical of ICF scenarios, such as very high temperatures and pressures.

The topic is both original and highly relevant within the fields of high-energy physics and ICF research, as it addresses a specific gap in current diagnostic methodologies. Conventional diagnostic instruments generally achieve temporal resolutions of around 50 picoseconds, which are insufficient for the rapid dynamics involved in ICF experiments. By implementing electron pulse dilation, the study introduces a novel approach that improves signal amplification and detection speed, significantly enhancing the measurement capabilities needed for such demanding experimental conditions.

Moreover, this research directly targets the limitations faced by existing diagnostic techniques in ICF, where the extreme conditions of the experiments necessitate instruments that can respond much faster. Traditional detectors struggle to provide the necessary temporal resolution to accurately characterize the dynamics occurring near stagnation, a critical phase in fusion processes. The development of the PD-PMT not only advances the state of temporal resolution in diagnostic instruments but also makes a valuable contribution to the broader field of ultrafast diagnostics. This innovation fills a crucial void in the available instrumentation for high-density physics experiments, paving the way for more precise investigations and a deeper understanding of the fundamental processes involved in inertial confinement fusion. By enhancing the capabilities of diagnostic tools in this way, the research fosters progress in the exploration of complex physical phenomena under extreme conditions.

The references are numerous and relevant.

Technical notes:

The paper is edited professionally. I did not detect any typos. The equations are nicely prepared.
All of the figures are good quality.

Summary:

In my overall opinion, this manuscript should be accepted in its present form.

Reviewer 3 Report

Comments and Suggestions for Authors

The paper provides a clear and compelling description of the design and purpose of the PD-PMT, which underlines the argument for its high-speed detection capabilities and improved temporal resolution. For instance, the explanation of converting signals into a free electron cloud is logically sound and technically plausible. The concept of utilizing electron pulse dilation in combination with magnetic imaging and time-resolved electron detection is a standout innovation. Traditional photomultipliers or time-of-flight detectors do not commonly employ such a combination, particularly at the precision discussed (sub-20 ps resolution).

The experimental investigation of the relationship between MCP gain and electron energy at the first collision provides a unique contribution. It delves into the foundational behavior of MCPs under specific conditions, offering insights into their performance characteristics that are rarely examined in such detail.

However, the paper could still accentuate its originality by providing comparisons with existing techniques like streak cameras or other high-speed photomultiplier systems. This would shed light on the specific advancements and unique contributions of the PD-PMT technology beyond its immediate experimental results.

In summary, while the research introduces original elements both in the PD-PMT’s design and function, further emphasis on its unique contributions and potential applications could bolster the perception of its originality. Explicit comparisons and real-world examples will strengthen its standing as a significant advancement in high-speed photodetection technology.

For the above reasons I consider that the article could be accepted by making minor corrections.
